# Graph Linearization Methods for Reasoning on Graphs with Large Language Models

## Abstract

Large language models have evolved to process multiple modalities beyond text, such as images and audio, which motivates us to explore how to effectively leverage them for graph reasoning tasks. The key question, therefore, is how to transform graphs into linear sequences of tokens—a process we term "graph linearization"—so that LLMs can handle graphs naturally. We consider that graphs should be linearized meaningfully to reflect certain properties of natural language text, such as local dependency and global alignment, in order to ease contemporary LLMs, trained on trillions of textual tokens, better understand graphs. To achieve this, we developed several graph linearization methods based on graph centrality and degeneracy. These methods are further enhanced using node relabeling techniques. The experimental results demonstrate the effectiveness of our methods compared to the random linearization baseline. Our work introduces novel graph representations suitable for LLMs, contributing to the potential integration of graph machine learning with the trend of multimodal processing using a unified transformer model.

## 1 Introduction

Transformer-based large pre-trained models have revolutionized machine learning research, demonstrating unprecedented performance across diverse data modalities and even a mixture of modalities, including image, audio, and text domains (Xu et al., 2023; Yin et al., 2023). In particular, large language models (LLMs) have shown promising results in arithmetic, symbolic, and logical reasoning tasks (Hendrycks et al., 2020). Despite their success, the adaptation for processing graphs—an ubiquitous data structure that encapsulates rich structural and relational information—remains a comparably emerging and underdeveloped research direction, even if it has recently been gaining attention (Ye et al., 2023; Fatemi et al., 2023; Wang et al., 2024). This asymmetry is due in large part to the inherent challenge of representing graphs as sequential tokens, in a manner conducive to the language modeling objectives typical of transformers, a challenge not encountered when dealing with the other modalities. This unique problem has encouraged us to investigate a key question: *How can we represent graphs as linear sequences of tokens for transformers in a suitable way?* We refer to this research endeavor as *Graph Linearization*.

Existing methods of using LLMs for graph machine learning tasks, such as graph reasoning and graph generation, represent entire graphs as either raw edge lists or natural language descriptions that adhere to adjacency matrices without any special treatment (Fatemi et al., 2023; Wang et al., 2024; Yao et al., 2024). For example, a natural language description of the star graph $S$ might be: "An undirected graph with nodes $a$, $b$, $c$, and $d$. Node $b$ is connected to $a$. Node $c$ is connected to $b$. Node $b$ is connected to $d$.", its equivalent edge list representation is: "[$(b, a)$, $(c, b)$, $(b, d)$]". Either of the linearized representations is then appended with a task-specific question to form an LLM query prompt, e.g., "Is there a cycle in this graph?". Other studies focus solely on node-level tasks (Zhao et al., 2023; Ye et al., 2023), centering the linearization around an ego-subgraph for a target node (Hamilton et al., 2017), where the neighboring graph structure and node features up to $k$-hop are described in the prompt. However, there is a lack of studies investigating how to maintain the integrity of graph structures while efficiently transforming them into sequences suitable for LLMs.

Our research addresses this limitation. By relying on edge list representations as exemplified above, we study the performance impact on LLMs of various methods to order the edges in the list and

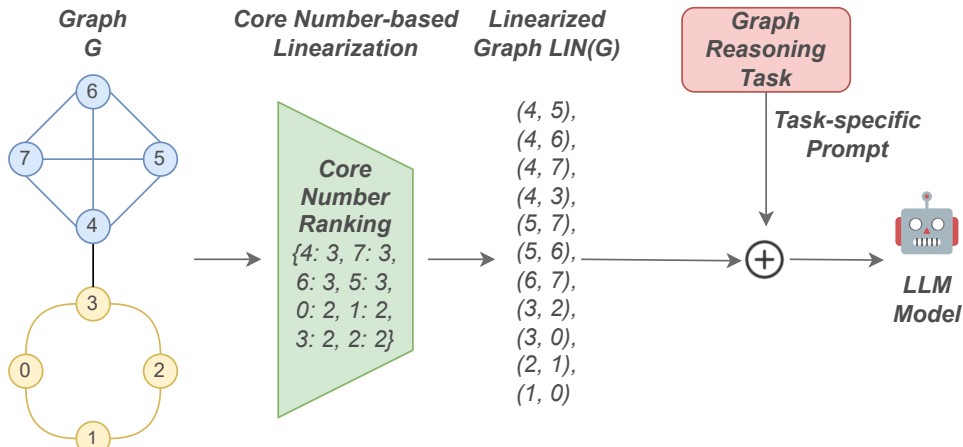

Figure 1: An example of *core number-based graph linearization* method. Given an input graph $G$, we rank its nodes based on their degree and then explore the edges in that order. The resulting linearized $LIN(G)$ graph is then combined with a task-specific prompt and passed into a LLM.

rename interchangeable node labels, as shown in Figure 1. We argue that if the linearization of graphs is conducted in a meaningful way, capturing properties similar to those found in natural language such as *local dependency* and *global alignment*, it will benefit contemporary LLMs by enhancing their ability to understand graphs, as they are trained on trillions of textual tokens.

To this end, we introduce two guiding principles for graph linearization: *local dependency* and *global alignment*. Local dependency refers to arranging edges so that structurally related components of the graph appear close together in the token sequence, increasing the likelihood that the model can use its contextual window effectively, much like how adjacent words in natural language share semantic relationships. Global alignment refers to consistently relabeling interchangeable node identifiers across examples so that structurally important nodes occupy similar token positions, mirroring the consistent placement of key words or phrases in natural language prompts.

Addressing this question helps identify LLM-suitable representations for graphs that potentially align with natural languages, unlocking new insights and applications in fields where graphs naturally represent data. Graph linearization paves the way for extending transformer models' capabilities to graph data and unifying various graph tasks across domains, serving as a fundamental step toward building successful large graph models. This also facilitates the integration of graph learning with the multi-modal processing trend and the development of cohesive AI systems using a unified transformer model.

In this work, we propose general graph linearization methods that leverage centrality and degeneracy measures, further enhanced through node relabeling to capture language-like properties. Our experiments cover both graph reasoning tasks and real-world classification datasets. We observe that our structured linearizations consistently outperform random orderings and achieve competitive performance compared to GNN baselines on real-world datasets. Our key findings are:

- **Ordering matters** – Structured graph linearizations improves the reasoning capabilities of LLMs.
- **Node relabeling helps** – Incorporating structural features into labels further enhances accuracy.
- **Task-specific orderings** – Node- and edge-based strategies excel in their respective tasks.

## 2 RELATED WORK

**Transformers for graph machine learning** Transformers (Vaswani et al., 2017) have been successfully applied in various domains beyond text, demonstrating both versatility and effectiveness (Devlin et al., 2018). For instance, the Vision Transformer (Dosovitskiy et al., 2020) has achieved remarkable performance in image classification tasks by treating images as sequences of patches,

marking a shift from traditional CNN-based approaches. Similarly, transformers have been used in speech recognition. Models like the Speech Transformer (Dong et al., 2018) apply self-attention mechanisms to process audio data as sequences, outperforming traditional RNN-based methods.

These successes have spurred interest in using modified transformers to replace the de facto GNN-based approaches for graph machine learning tasks. Notably, Graphormer (Ying et al., 2021) enables transformer to effectively capture the dependencies and relationships within a graph. It achieves this by integrating node centrality encoding and attention biases that account for the spatial distance between nodes. Graph Transformer (GT) (Dwivedi & Bresson, 2020) generalizes the transformer architecture for graph representation learning. GT introduces the concept of relative positional encodings to account for the pairwise distances between nodes in a graph. This approach allows the model to learn rich node representations that capture both local and global graph structures.

In contrast to the above works, Kim et al. (2022) show that by treating all nodes and edges as independent tokens and inputting them into a standard Transformer encoder without any graph-specific modifications, notable outcomes can be achieved both theoretically and practically. Results on molecular graphs for quantum chemical property prediction show that this approach outperforms all GNN baselines and achieves competitive performance compared to graph Transformer variants. Despite not applying any structural alterations to the Transformer, this approach still requires sophisticated token-wise node and edge embeddings to explicitly represent the connectivity structure.

**LLMs for graph reasoning** Following the recent success of LLMs in tasks beyond language processing (Hendrycks et al., 2020), several studies have explored the capacity of off-the-shelf LLMs for *graph reasoning*. While there is no clear consensus on the specific tasks, models are tested on understanding basic topological properties, such as graph size, node degree and connectivity, which form the foundation for a deeper understanding of graph structures (Zhang et al., 2023b). Using various prompting methods, these studies show that LLMs, even without fine-tuning, demonstrate preliminary graph reasoning abilities.

Several studies have evaluated LLMs for graph reasoning at both the node and graph levels. For example, NLGraph (Wang et al., 2024) covers eight graph reasoning tasks of varying complexity, ranging from simple tasks like connectivity and shortest path to complex problems like maximum flow and simulating graph neural networks. This work also proposes two graph-specific prompting methods that achieve notable performance improvements. GraphQA (Fatemi et al., 2023) focuses on relatively simple tasks to measure the performance of pre-trained LLMs in edge existence, node degree, node count, edge count, connected nodes, and cycle checks. It shows that larger models generally perform better on graph reasoning, with graphs generated synthetically using various graph generators. Similar works include various studies that explore graph reasoning using different LLMs, prompting techniques, graph tasks, domains, and evaluation approaches (Chen et al., 2023; Guo et al., 2023; Zhang et al., 2023a; Hu et al., 2023; Huang et al., 2024; Liu & Wu, 2023; Das et al., 2023; Yuan et al., 2024; Wu et al., 2024; Skianis et al., 2024).

Another line of research criticizes the above approach, arguing that solely using prompt engineering or in-context learning with frozen LLMs hinders achieving top performance in downstream graph tasks. Therefore, instruction-tuning or fine-tuning is necessary. The work of Ye et al. (2023) preliminarily confirms this on the multi-class node classification task. A prompt template is designed to describe both the neighbor graph structure and node features centered around a target node up to the 3-hop level. Similarly, Zhao et al. (2023) draw inspiration from linguistic syntax trees. For a target node, the work converts its ego-subgraph into a graph syntax tree with branches describing the neighborhood's "label" and "feature", which are then encoded as text in the prompt. Results show that instruction-tuning performs much better than in-context learning, and is on par with GNN-based models. Similar conclusions can be observed in recent studies on graph-level reasoning settings (Luo et al., 2024). Finally, Perozzi et al. (2024) introduce GraphToken, which trains an encoder to create continuous representations, rather than converting graphs into text tokens.

While these approaches demonstrate strong performance, they rely on fine-tuning or additional components beyond the base LLM. In contrast, our method requires no training or parameter updates. We focus on input-level graph linearization techniques that make graphs more interpretable to frozen LLMs, maximizing their utility on graph reasoning tasks without any model modification.

**Linearization for specific graphs** In deriving an ordering for graphs, topological sorting in graph theory examines the linear ordering of directed acyclic graphs, such that for every directed edge

$(u, v)$, $u$ precedes $v$ in the ordering. However, such graph traversal is node-centric, making edge information not encoded.

In other domains involving specific types of graphs, such as discourse graphs—a directed weakly connected graph reflecting discourse structure—nodes represent utterances, and edges represent discourse relations (e.g., elaboration, clarification, completion) within a conversation (Rennard et al., 2024). The work of Chernyavskiy et al. (2024) proposes a linearization method for discourse graphs that arranges utterances chronologically, assigning unique identifiers to speakers, utterances, and addressees. It incorporates discourse relations and sentiment tokens to generate a structured sequence, using special tokens for clarity. This structured sequence is then used to train a BART (Lewis et al., 2020) for dialogue generation. Similarly, Abstract Meaning Representation (AMR) uses directed acyclic graphs to provide a structured semantic representation of language, incorporating semantic roles with annotated arguments and values where nodes represent concepts and edges represent semantic relations (Banarescu et al., 2013). AMR corpora are usually linearized using the PENMAN-based notation (Patten, 1993) as in the work of (Ribeiro et al., 2021) and (Hoyle et al., 2021) to fine-tune pre-trained language models to perform graph-to-text generation. For citation networks, Guo et al. (2023) have explored the Graph Modeling Language (GML) and Graph Markup Language (GraphML) for graph representation (Himsolt, 1997; Brandes et al., 2013). GML is a simple, human-readable format, while GraphML is XML-based and offers extensibility for complex applications.

**The scope of our work**. Unlike the above linearization methods limited to specific types of graphs, where linearization can be naturally derived to some extent, we focus on general graphs. Furthermore, unlike previous works using LLMs for reasoning, where edge lists are directly leveraged without special treatment, we introduce various linearization methods for ordering the edges in the list and renaming interchangeable node labels to make them suitable for LLMs. Although our work involves mostly graph reasoning experiments, our graph linearization methods are general and applicable to various scenarios. This allows for the effective transformation of graph structures into sequences suitable for language models and has the potential to improve performance in variety of graph tasks, with or without fine-tuning.

## 3 GRAPH LINEARIZATION

This section describes our graph linearization approaches, emphasizing the use of graph features to enhance graph reasoning with LLMs.

Generally speaking, we define graph linearization as the process of representing graphs as linear sequences of tokens. In this work, we aim to identify the linearization approaches that will benefit LLMs by enhancing their ability to understand graphs. We argue that linearized graphs, represented as sequences of textual tokens, should capture properties similar to those in natural language, given the fact that LLMs are pre-trained on trillions of textual tokens. Such properties should include local dependency and global alignment.

By **local dependency** we refer to the capacity to predict the subsequent (or missing) token based on the preceding (or surrounding) context within the token sequence of a single linearized graph. This property is analogous to the foundational *distributional hypothesis* of language (Joos, 1950; Harris, 1954; Firth, 1957), which asserts that words occurring in similar contexts tend to exhibit similar meanings or functions. The hypothesis suggests that, upon encountering an unfamiliar word, its meaning can be inferred from the contexts in which it appears. In the context of graphs, this implies that grouping structurally related components enhances the likelihood that an LLM's contextual window will capture meaningful connectivity patterns.

Similarly, **global alignment** pertains to the alignment of token sequences across different linearized graphs, ensuring that corresponding tokens align consistently across examples. This property captures the overarching structure of the sequences, reflecting the predictable flow of natural language, wherein common words tend to occupy consistent positions within a sequence. Thus, global alignment can be facilitated by consistently initiating the linearization from structurally important nodes, such as starting from the highest-degree node, and by ensuring that these nodes are consistently labeled as index 0 across graphs.

To empirically evaluate these properties, we consider three distinct categories of measures: *Degree* (local connectivity), *PageRank* (global influence), and *Core number / degeneracy* (membership in dense substructures). These measures are applied in both node-centric linearizations, where incident edges are emitted while scanning nodes in rank order, and in an edge-centric variant, by applying the same pipeline to the line graph (L{G}).

## 3.1 IMPLEMENTATION

Our approach to capitalizing on the local dependency property involves the following steps. Given a graph $G$, we initially rank the nodes by the centrality and degeneracy measures described previously. Then, we begin exploring the nodes by descending order and list the edges connected to it, arranging them in a random order. Each edge is represented as a node pair. In the case where two or more nodes share an equal value, the order is selected randomly. After the ordering process has concluded, each edge list constitutes a sequence of tokens following a descending order of node importance.

In addition to linearization methods, node relabeling is employed as a means to attempt the attainment of the global alignment property. Specifically, node relabeling introduces an additional step to our procedure. After ranking the nodes, their original labels are replaced with their respective positions in the ranking. Consequently, the node with index 0 corresponds to the one with the highest core number, and so forth. This approach may prove advantageous for the LLM by ensuring a consistent association between node indices and their respective importance properties.

Finally, we conducted experiments in which the edges were ordered directly, rather than the nodes. This allows our linearization to directly capture relationships between edges, which can be essential for understanding complex graph structures. To achieve this, each graph was transformed into its corresponding linegraph representation. A linegraph $L(G)$ of a graph $G$ is the graph where each edge of $G$ is replaced by a node, and where two edges of $G$ are connected in $L(G)$ if they are incident in $G$. Subsequently, the previously described processes were applied directly to $L(G)$.

## 4 EXPERIMENTAL SETUP

In this section, we present a comprehensive overview of the experimental setup, detailing the methodologies, resources, and evaluation frameworks employed in our experiments. We first describe experiments conducted on synthetic datasets designed for graph reasoning tasks, followed by experiments on real-world protein-based datasets for graph classification tasks.

### 4.1 SYNTHETIC DATASETS

### 4.2 DATASETS

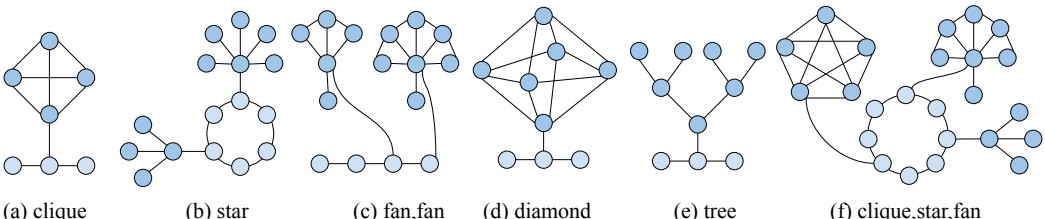

(a) clique      (b) star      (c) fan,fan    (d) diamond      (e) tree      (f) clique,star,fan

Figure 2: Overview of GraphWave synthetic dataset.

**GraphWave.** First, we constructed a synthetic graph dataset using `GraphWave` (Donnat et al., 2018). This graph generator was originally developed for controlled experimentation and evaluation of node embedding techniques and measurement of structural equivalence on graphs with known network motifs. Motifs, in the context of network science, are sub-graphs that repeat themselves either within the same graph or across different graphs. For our analysis, these structurally similar sub-graphs enable us to construct structure-related tasks, that allow us to assess the ability of language models to analyze and infer structural features within these graphs.

The generator operates by sequentially constructing a *base* graph, that follows either a cycle or chain structure, and then attaching a number of *motifs* that follow predetermined shapes—cliques, stars, fans, diamonds, and trees. To cover a wider variety of structural complexities, we also include all combinations of two shapes, along with all combinations of three shapes with unique shapes per triplet. Example graphs can be found in Figure 2.

For each combination of shapes, we generated 100 graphs, leading to a total of 3000 graphs. This ensures sufficient variance in graph sizes and in the combinations of base and motif sub-graphs. The number of nodes in each graph shape is selected randomly, with the following constraints: base (3-21 nodes); clique, fan, and star (4-11 nodes); diamond (6 nodes); and tree (perfect binary trees with 3-6 levels). The generated dataset contains an average of 32.33 nodes and 43.72 edges per graph.

**GraphQA.** Furthermore, we included `GraphQA` (Fatemi et al., 2023), which is widely used to evaluate the graph reasoning capabilities of LLMs. Similar to GraphWave, the dataset consists of randomly generated graphs derived from various graph generators, including Erdős–Rényi graphs (Erdős & Rényi, 1959), scale-free networks (SFN) (Barabási & Albert, 1999), the Barabási–Albert model (BA) (Albert & Barabási, 2002), and the stochastic block model (SBM) (Holland et al., 1983), along with star, path, and complete graph generators. A total of 500 graphs were sampled for ER, BA, SFN, and SBM models, whereas 100 graphs were sampled for path, complete, and star graphs due to their lower structural variability. All generated graphs contained between 5 and 20 nodes.

Table 1: Accuracy scores for all tasks on the **GraphWave** dataset using **Llama 3 8B**, including the overall average. Each task compares two node labeling schemes in zero-shot / one-shot prompting against the random `Baseline`. The linearization names represent the node ordering methods used (CoreNumber, Degree, or PageRank), while *'LG{\*}'* indicates graphs were transformed to the corresponding linegraph beforehand. Underlined scores denote the best-performing linearization method for each task, labeling, and X-shot combination; bold indicates task-wise best.

| | Node Counting | Max Degree | Node Degree | Edge Existence | Diameter | Shortest Path | Path Existence | Motifs' Shape | Average |
|---|---|---|---|---|---|---|---|---|---|
| *Random Labeling* | | | | | | | | | |
| CoreNumber | 25.97 / 34.98 | 17.47 / 17.37 | 58.53 / 56.79 | 51.83 / 55.29 | 8.9 / 11.87 | 24.57 / 17.84 | 85.17 / 66.16 | 45.8 / 64.07 | 39.78 / 40.55 |
| Degree | 28.0 / 36.98 | 27.63 / 27.14 | 60.83 / 52.22 | 47.6 / 53.72 | 7.97 / 10.17 | 27.8 / 15.81 | 84.63 / 69.12 | 48.47 / 64.8 | 41.62 / 41.24 |
| PageRank | 28.81 / 39.18 | 24.10 / 23.57 | 59.4 / 56.42 | 44.9 / 47.52 | 8.83 / 11.27 | 27.37 / 16.34 | 84.63 / 72.69 | 44.93 / 65.73 | 40.37 / 41.59 |
| LG{CoreNumber} | 21.9 / 27.38 | 19.00 / 16.47 | 46.23 / 42.68 | 52.47 / 50.12 | 8.73 / 12.0 | 23.03 / 16.21 | 83.87 / 64.22 | 44.4 / 63.13 | 37.45 / 36.53 |
| LG{Degree} | 20.4 / 26.51 | 27.63 / 18.87 | 48.07 / 47.92 | 55.47 / 52.65 | 8.7 / 11.94 | 26.3 / 17.64 | 84.8 / 62.29 | 44.33 / 61.17 | 39.46 / 37.37 |
| LG{PageRank} | 29.1 / 27.28 | 18.83 / 17.97 | 46.8 / 40.61 | 47.9 / 49.02 | 8.47 / 12.14 | 24.73 / 16.04 | 87.07 / 70.52 | 42.4 / 60.0 | 38.16 / 36.7 |
| *Node Relabeling* | | | | | | | | | |
| CoreNumber | 28.65 / 36.05 | 14.57 / 16.17 | 58.3 / 61.32 | 59.43 / 59.89 | 9.7 / 10.54 | 27.53 / 18.57 | **88.0** / 72.92 | 44.3 / 64.57 | 41.31 / 42.5 |
| Degree | 31.44 / **43.85** | **29.40** / 32.48 | 62.93 / 56.79 | 58.5 / 55.09 | **11.1** / 11.24 | 30.3 / 16.31 | 82.0 / 72.79 | 48.37 / **76.3** | **44.26** / **45.61** |
| PageRank | **34.35** / 38.01 | 26.07 / 25.24 | 65.37 / 56.92 | 50.83 / 46.08 | 10.47 / 11.17 | 32.33 / 17.67 | 82.13 / 72.89 | 47.6 / 68.63 | 43.64 / 42.08 |
| LG{CoreNumber} | 15.64 / 20.34 | 13.33 / 8.67 | 56.23 / 48.15 | 63.97 / 55.55 | 8.5 / **12.5** | 27.6 / 17.67 | 85.83 / 70.66 | 45.0 / 50.9 | 39.51 / 35.56 |
| LG{Degree} | 23.59 / 31.94 | 19.83 / 16.11 | 48.9 / 47.85 | 60.03 / **63.05** | 10.1 / 12.14 | 29.63 / 17.31 | 84.8 / 70.36 | 41.07 / 54.23 | 39.74 / 39.12 |
| LG{PageRank} | 27.57 / 37.45 | 17.03 / 18.64 | 51.8 / 48.45 | 54.37 / 51.15 | 9.73 / 12.34 | 26.6 / 15.41 | 85.47 / 71.72 | 39.4 / 51.77 | 39.0 / 38.37 |
| **`Baseline`** | 32.46 / 34.6 | 15.13 / 12.94 | 39.36 / 45.99 | 36.99 / 53.25 | 7.99 / 12.38 | 22.73 / 16.57 | 86.98 / 68.05 | 43.53 / 66.8 | 34.86 / 34.86 |

### 4.2.1 TASKS

Our experiments encompass a series of graph reasoning tasks, which can be broadly categorized into classification-based and numerical tasks. Numerical tasks, ranging in difficulty, require the model to produce a numerical output, either through structural computation or counting-based inference. This combination of tasks allows us to comprehensively evaluate LLMs' understanding of structural features and examine how different edge list orderings affect their performance.

A fundamental task is `Node Counting`, where the LLM estimates the number of nodes. In `Node Degree` calculation, the LLM determines the degree of a node. A more advanced variant, `Maximum Degree` calculation, requires the LLM to internally calculate the degree of all nodes and then identify the maximum among them.

Beyond node-related tasks, we assess the ability to infer relational properties. In `Edge Existence` and `Path Existence` tasks the LLM is given a randomly selected pair of nodes and must determine whether an edge or a connecting path exists between them, respectively. In the `Shortest Path` task, the model must compute the length of the shortest path between two given nodes, requiring a deeper understanding of graph connectivity. The `Diameter Estimation` task requires the model to determine the longest shortest path in the graph, showcasing global graph structure understanding.

Finally, we evaluated `Motifs' Shape` classification, a dataset-specific task leveraging Graph-Wave's embedded structures, where the LLM is given definitions of the five motif types and asked is to identify which is present.

In every prompt, a node $v$ is represented by an incremental integer, while an edge between nodes $v$ and $u$ is denoted by the bracketed pair $(v, u)$. An edge list is expressed as a sequence of edges, sorted according to the scheme used in each linearization method. We tested both zero-shot and one-shot approaches, where a randomly selected graph from the dataset was used consistently as the one-shot example across all experiments. The prompt templates are provided in Appendix D.

### 4.2.2 LLMs

We used the 8B parameter Llama 3 Instruct (Dubey et al., 2024) with a temperature of $1e-3$ and a sampling parameter of $1e-1$ for more deterministic outputs to assess sensitivity to our linearization methods. Experiments were conducted on an NVIDIA A5000. Further experiments with different model sizes and families, including the Llama 3 70B and Qwen 2.5 14B-1M (Yang et al., 2025), are provided in Appendix A.

### 4.2.3 BASELINES

For our comparisons, we consider a random baseline. This baseline involves a fully random ordering of the edge list, where edges are arranged without following any inherent scheme. To further eliminate structural biases, we also randomly shuffle the node labels. This baseline is founded on the fact that we are working with general graphs, where default labels or ordering are neither pre-determined nor necessarily provide meaningful information in real-world applications. In addition, to mitigate the risk of skewed results, we applied five different random orderings and averaged their performance. Our random ordering can be considered comparable to prior studies, which tend to preserve the inherent structure of the generator.

Table 2: Accuracy scores for all tasks on the **GraphQA** dataset using **Llama 3 8B**, , including the overall average. Notations remain the same as in Table 1.

| | Node Counting | Max Degree | Node Degree | Edge Existence | Diameter | Shortest Path | Path Existence ‖ | Average |
|---|---|---|---|---|---|---|---|---|
| ***Random Labels*** | | | | | | | | |
| CoreNumber | 60.77 / 24.28 | 15.88 / 18.67 | 54.44 / 37.23 | 70.68 / 61.59 | 3.28 / **18.53** | 49.21 / **57.25** | 95.51 / 98.39 | 49.97 / 45.13 |
| Degree | 62.18 / 28.91 | 25.6 / 29.75 | 54.99 / 29.4 | 68.02 / 58.6 | 3.39 / 14.82 | **50.24** / 53.34 | 94.39 / 98.71 | 51.26 / 44.79 |
| PageRank | 62.12 / 28.31 | 26.03 / 28.94 | 56.23 / 28.05 | 68.39 / 59.26 | 3.16 / 15.25 | 29.34 / 47.3 | 94.51 / 98.68 | 48.54 / 43.68 |
| LG{CoreNumber} | 61.14 / 23.94 | 10.81 / 25.89 | 49.09 / 36.65 | 69.26 / 58.83 | 2.9 / 16.97 | 42.08 / 42.35 | 95.92 / 98.71 | 47.31 / 43.33 |
| LG{Degree} | 59.53 / 27.22 | 12.97 / 28.25 | 48.46 / 37.69 | 68.25 / 60.39 | 2.7 / 16.34 | 23.99 / 43.1 | 95.2 / **98.82** | 44.44 / 44.54 |
| LG{PageRank} | 60.66 / 26.24 | 11.59 / 28.65 | 47.51 / 38.43 | 69.49 / 62.66 | 2.67 / 17.43 | 40.78 / 45.6 | 94.31 / 98.79 | 46.72 / 45.4 |
| | | | | | | | | |
| ***Node Relabeling*** | | | | | | | | |
| CoreNumber | 61.06 / 34.38 | 22.09 / 30.29 | 51.94 / 30.49 | 72.02 / 70.17 | 2.36 / 14.41 | 46.28 / 52.85 | 94.79 / 98.56 | 50.08 / 47.31 |
| Degree | 65.31 / **45.48** | 39.11 / **44.3** | 54.76 / 29.2 | 70.18 / **74.6** | 2.93 / 14.3 | 46.22 / 52.01 | 96.89 / 98.76 | **53.63** / **51.24** |
| PageRank | 64.71 / 44.97 | **40.72** / 43.15 | **58.77** / 28.65 | 69.97 / 70.83 | **3.57** / 14.13 | 26.17 / 49.54 | **97.04** / **98.82** | 51.56 / 50.01 |
| LG{CoreNumber} | 63.56 / 32.54 | 22.69 / 33.49 | 47.48 / 29.14 | **72.91** / 59.15 | 1.93 / 15.1 | 24.33 / 43.84 | 96.52 / **98.82** | 47.06 / 44.58 |
| LG{Degree} | 64.77 / 35.07 | 27.87 / 35.27 | 47.2 / **39.18** | 70.46 / 59.26 | 3.19 / 15.13 | 42.45 / 46.72 | 96.06 / 98.76 | 50.29 / 47.06 |
| LG{PageRank} | **69.37** / 32.42 | 26.37 / 36.02 | 49.5 / 38.9 | 70.84 / 63.98 | 3.13 / 15.19 | 41.53 / 47.44 | 95.48 / 97.96 | 50.89 / 47.42 |
| | | | | | | | | |
| **Baseline** | 66.28 / 27.38 | 9.28 / 20.37 | 49.78 / 35.86 | 67.02 / 58.99 | 2.94 / 14.98 | 37.7 / 44.55 | 95.83 / 98.51 | 46.98 / 42.95 |

### 4.2.4 EVALUATION

We use exact accuracy to compare our methods, measuring the ratio of correct predictions as $\frac{1}{n} \sum_{i=1}^{n} I(y_i = \hat{y}_i)$, where $n$ is the number of graphs, $y_i$ the correct answer, and $\hat{y}_i$ the LLM's response. For numerical tasks, we consider a result accurate only if it is an exact match. For the motifs' shape classification task, accuracy reflects the total across all shapes, requiring the predicted shape to appear at least once in the graph.

### 4.3 REAL-WORLD DATASETS

To further assess the effectiveness of our graph linearization methods beyond synthetic datasets, we evaluate them on two widely used real-world graph binary classification datasets from the TUDataset collection (Morris et al., 2020): **AIDS** and **MUTAG**. These datasets consist of molecular graphs, where nodes represent atoms and edges represent chemical bonds, offering a more challenging setting. For this experiment, we employ the **Qwen 2.5 14B - 1M** language model, which supports

extended context lengths and can encode full graph sequences as input. Our aim is to determine whether zero-shot LLMs, when given only linearized graph representations and task prompts, can generalize to graph classification tasks without any parameter tuning.

To benchmark performance, we compare against two standard **GCN** models: one using only trainable node embeddings and another incorporating the original node features provided by the dataset. These models are trained end-to-end on 80% of the data, with evaluation on the remaining 20% test split. In contrast, the LLM is evaluated *only* on the test set using zero-shot prompting, without any access to the training data. This separation allows us to measure the generalization ability of frozen LLMs under realistic constraints and to isolate the impact of linearization in the absence of task-specific training.

Table 3: Accuracy scores for TUDatasets (Morris et al., 2020) AIDS and MUTAG using **Qwen 2.5 14B - 1M**. We compare against two basic GCN models, one where we ignore the node features and instead each node is assigned a trainable embedding vector and one where the node features are included. The train split is 80%, the LLM is evaluated over the test split only with a zero-shot setting. Underlined scores denote best performing linearization; bold indicates dataset-wise best.

|  | AIDS | MUTAG |
|---|---|---|
| **Random Labels** | | |
| CoreNumber | 55.14 | 66.42 |
| Degree | 54.64 | 65.79 |
| PageRank | 46.62 | 60.53 |
| LG{CoreNumber} | 51.63 | 66.42 |
| LG{Degree} | 59.92 | 63.16 |
| LG{PageRank} | 59.4 | 65.79 |
| **Node Relabeling** | | |
| CoreNumber | 53.13 | 65.79 |
| Degree | 58.4 | 65.79 |
| PageRank | 50.38 | 68.42 |
| LG{CoreNumber} | 51.63 | 65.79 |
| LG{Degree} | 76.19 | 65.79 |
| LG{PageRank} | 68.42 | 63.16 |
| **Random Ordering** | 40.5 | 65.79 |
| **Default Ordering** | 32.08 | 63.16 |
| **GCN** | **80.0** | 67.56 |
| **GCN-NodeFeatures** | 79.75 | **84.21** |

## 5 EXPERIMENTAL RESULTS ANALYSIS

The performance of our methods is presented in Tables 1 and 2, which report results on the Graph-Wave and GraphQA datasets with the Llama 3 8B model. Results on real-world datasets with the Qwen2.5 14B model are provided in Table 3. The results are organized into three groups: one where node labels in the linearized graphs are randomly assigned, ensuring a fair comparison since, in real-world graphs, labels might be arbitrary; another where node labels are reindexed according to each method, as described in Section 3; and finally, a comparison against the baselines. The performance related to the pseudo-random (default) node labels originally provided by synthetic graph generators is discussed in Appendix A.

Overall, across both synthetic datasets, our linearization methods consistently outperform the random baseline, highlighting the critical role of graph linearization, as evident in the average performance and across multiple individual tasks. Notably, on the GraphQA dataset, the combination of degree-based ordering and node relabeling improves performance by approximately 35% on the maximum node degree estimation task and by around 13% on the shortest path task. Similarly, on the GraphWave dataset, the combination of degree-based ordering and node relabeling enhances edge existence performance by roughly 26%. Comparing random and node relabeling reveals that

ordering alone offers significant improvements over the baseline, while the additional information from structured relabeling further enhances accuracy on nearly all tasks. Among all tasks, diameter estimation is the most challenging across both datasets, with consistently low performance, indicating LLMs struggle to infer global graph properties at this level of complexity.

Linegraph-based methods (LG{*})—where edges are reinterpreted as nodes—highlight the importance of edge-to-edge relationships. While their overall average score is lower, they generally perform better in edge-based tasks, such as edge existence and path reasoning, by capturing interdependencies that might be less evident in traditional node-focused representations. These findings suggest that a more suitable linearization approach may be necessary to fully exploit the benefits of the linegraph transformation.

Similarly, CoreNumber-based methods achieve better performance in edge-centric tasks, which can be attributed to its ability to capture the structural cohesiveness of a graph. By emphasizing nodes embedded in densely connected subgraphs, core number ordering effectively preserves key connectivity patterns, making it particularly advantageous for reasoning about edge relationships. In contrast, while Degree- and PageRank-based orderings demonstrate the most consistent performance across various tasks, their strengths are more pronounced in node-related tasks.

When moving from zero-shot to one-shot setting, we notice a performance loss on binary classification tasks like edge and path existence. This decline may result from the model's reliance on a single graph example, which does not fully capture the complexity and diversity of the dataset. However, despite this drop, one-shot prompting remains effective for more complex tasks.

The results in the real-world datasets demonstrate that structured graph linearization methods significantly improve the zero-shot graph classification performance of a frozen LLM (Qwen 2.5 14B) on real-world datasets, compared to both random and default orderings. Notably, the linegraph-based linearization with degree ordering and node relabeling (LG{Degree}) achieves the best LLM performance on the AIDS dataset (76.19%), outperforming all other linearizations and approaching the GCN baseline (80.0%). On MUTAG, the best LLM result (68.42%) is achieved by PageRank with node relabeling, slightly exceeding the GCN (67.56%) and only lagging behind the GCN variant that uses node features (84.21%). These findings support our core claim: that carefully designed graph linearizations—without any training or fine-tuning—can make frozen LLMs competitive on graph-based tasks. Although LLMs have not yet matched the performance of fully trained GCNs in all scenarios, our approach significantly improves their performance through prompt engineering alone, highlighting its practicality over traditional GNNs.

## 6 CONCLUSION

This work addresses a fundamental challenge in using LLMs for graph-based reasoning: how to represent graph structures as linear textual token sequences in a way that aligns with the models' training on natural language. We explore a family of graph linearization methods that embed two key properties—*local dependency* and *global alignment*—directly into the sequence structure. Local dependency is encouraged by ordering edges based on node-centric measures such as degree, core number, and PageRank, allowing nearby tokens to reflect structurally related regions of the graph. Global alignment is introduced via node relabeling, ensuring that important nodes are consistently assigned the same token indices across examples, thereby aligning the positional priors LLMs rely on.

Our experiments show that even without fine-tuning, LLMs like LLaMA 3 and Qwen 2.5 benefit from these structural cues: performance on a range of graph reasoning tasks improves significantly over random orderings. Results on real-world datasets further demonstrate that these input-level strategies can generalize beyond synthetic graphs.

Unlike prior methods that adapt the model through training, we show that effective prompt design—when grounded in graph structure—can unlock latent capabilities of LLMs for graph tasks. By encoding structural bias into the input, our approach provides a simple and general path toward integrating graphs into the LLM ecosystem, with no architectural changes or training required.

## REPRODUCIBILITY STATEMENT

All experiments in this work are fully reproducible. Our study uses standard graph algorithms such as k-core decomposition, degree computation, and line graph construction, implemented with publicly available libraries. All LLM prompting parameters are described in Section 4.2.2, the exact prompts used are provided in Section D, and all LLMs used are publicly available.

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

## A ADDITIONAL EXPERIMENTS

**Pseudo-random linearization.** We also investigated the performance of utilizing the edge ordering directly provided by the graph generator. Most non-random graph generators create graphs procedurally, inherently embedding structural information within the edge list order. For instance, in GraphWave, the process begins with a base graph and subsequently attaches motifs, making it easier to distinguish between different structures. We hypothesize that this embedded structural knowledge will enhance task performance and boost the LLM's capabilities. However, generating such structure-aware edge lists requires an understanding of the graph construction process, which may not be feasible for real-world applications involving larger and more complex graphs.

The results of both datasets are presented in Table 4. For the GraphWave dataset, the default edge ordering shows mixed performance compared to the random ordering baseline. When combined with structured edge ordering (Table 1), accuracy improves consistently across tasks, except for path existence. When comparing default labeling with structured node labeling, performance generally improves, though default labeling remains competitive in certain tasks. For the GraphQA dataset, the default edge ordering performs significantly better than the random ordering across all tasks except path existence. In this case, the default ordering proves to be particularly robust, making it challenging for structured edge ordering to achieve higher accuracy. Even compared to structured edge ordering (Table 2), default ordering often maintains a performance advantage, highlighting its effectiveness in this dataset.

**LLM Family Variation with Extended Context.** To further examine the influence of model architecture and extended context capacity, we evaluated the performance of Qwen 2.5 14B-1M (Yang et al., 2025), a large-context language model from a distinct model family capable of processing input sequences of up to 1 million tokens. This evaluation allows us to assess whether architectural differences impact performance on graph-based reasoning tasks. Table 6 reports the results obtained for the same selection of tasks previously used in Table 5, enabling a direct comparison across models with varying capacities.

Table 4: Accuracy scores for all tasks on the **GraphWave** (top) and **GraphQA** (bottom) datasets using **Llama 3 8B**. For each task, we compare the default labeling scheme, as provided by the corresponding graph generator, against the default order the edges have been generated. Notations remain the same as in Table 1.

| | Node Counting | Max Degree | Node Degree | Edge Existence | Diameter | Shortest Path | Path Existence | Motifs' Shape |
|---|---|---|---|---|---|---|---|---|
| **Default Labels** | | | | | | | | |
| CoreNumber | 25.06 / 37.41 | 17.87 / 16.67 | 60.9 / **62.45** | 54.43 / **57.69** | 7.93 / 11.74 | 29.97 / **20.74** | 83.1 / 70.32 | 43.43 / 63.23 |
| Degree | 32.66 / **51.18** | **28.37** / 26.24 | 60.77 / 57.79 | 55.5 / 49.22 | 8.97 / 11.54 | **31.7** / 15.97 | 78.9 / 67.36 | 47.8 / 61.4 |
| PageRank | 35.77 / 49.38 | 24.40 / **26.88** | **63.43** / 56.42 | 49.4 / 53.08 | 8.63 / 11.5 | 31.33 / 18.24 | 81.3 / 73.99 | 48.03 / **65.27** |
| LG{CoreNumber} | 19.79 / 22.21 | 21.60 / 13.40 | 55.37 / 47.75 | **59.4** / 53.35 | 10.07 / 11.9 | 27.03 / 19.74 | 79.4 / 73.06 | 44.43 / 56.23 |
| LG{Degree} | 24.6 / 32.21 | 23.23 / 17.84 | 49.33 / 41.35 | 56.4 / 52.72 | **10.67** / 11.04 | 29.33 / 19.47 | 79.67 / 71.76 | 43.87 / 52.73 |
| LG{PageRank} | 34.83 / 41.78 | 17.90 / 19.14 | 54.3 / 50.18 | 48.9 / 52.15 | 8.77 / **12.17** | 27.63 / 16.47 | 79.77 / 76.59 | 43.07 / 48.63 |
| Default Ordering | **36.75** / 44.38 | 9.77 / 10.17 | 47.83 / 41.18 | 37.23 / 47.48 | 7.3 / 11.5 | 30.3 / 16.67 | **85.5** / **82.79** | **54.63** / 55.73 |

| | Node Counting | Max Degree | Node Degree | Edge Existence | Diameter | Shortest Path | Path Existence |
|---|---|---|---|---|---|---|---|
| **Default Labels** | | | | | | | |
| CoreNumber | 61.23 / 38.52 | 22.4 / 34.35 | 52.72 / **44.51** | 72.19 / 68.07 | 1.58 / 9.29 | **46.65** / 58.57 | 94.62 / 97.55 |
| Degree | 69.31 / **56.88** | 44.32 / **52.19** | 54.53 / 26.64 | 70.23 / 70.4 | 2.36 / 12.31 | 25.48 / 54.2 | 97.12 / 98.79 |
| PageRank | 69.03 / 53.97 | 44.23 / 52.1 | 54.56 / 33.8 | 69.66 / **71.66** | 2.88 / 11.94 | 45.41 / 50.86 | 96.98 / 98.79 |
| LG{CoreNumber} | 67.73 / 38.38 | 22.66 / 30.09 | 47.63 / 33.75 | 71.87 / 61.71 | 1.27 / 11.68 | 40.44 / 47.07 | 96.69 / 98.79 |
| LG{Degree} | 66.9 / 42.84 | 26.43 / 34.75 | 46.85 / 38.67 | 71.99 / 62.49 | 2.3 / 14.38 | 41.39 / 46.38 | 95.8 / **98.82** |
| LG{PageRank} | **70.87** / 37.46 | 24.82 / 35.13 | 49.5 / 37.83 | **72.36** / 58.92 | 2.24 / 16.4 | 41.53 / 47.61 | 95.17 / 97.93 |
| Default Ordering | 68.43 / 51.06 | **52.78** / 48.48 | **56.08** / 33.08 | 67.33 / 71.06 | **4.43** / **30.72** | 28.5 / 41.94 | **98.25** / 95.51 |

Table 5: Accuracy scores for a subset of tasks on the **GraphWave** dataset using **Llama 3 70B** as an ablation study. Notations remain the same as in Table 1.

| | Node Counting | Node Degree | Diameter | Motifs' Shape |
|---|---|---|---|---|
| **Random Labels** | | | | |
| CoreNumber | 76.47 / 75.29 | 72.47 / 63.75 | 3.97 / 4.07 | **59.1** / **61.43** |
| Degree | 78.2 / 80.96 | 73.93 / 63.69 | 2.43 / 4.57 | 58.07 / 61.4 |
| PageRank | 76.1 / 78.09 | 74.0 / 63.39 | 2.07 / 3.17 | 54.33 / 54.6 |
| LG{CoreNumber} | 77.8 / 77.36 | 68.83 / 57.02 | 4.77 / 6.97 | 46.8 / 56.23 |
| LG{Degree} | 78.9 / 81.16 | 72.23 / 62.65 | 4.57 / 7.5 | 51.5 / 54.2 |
| LG{PageRank} | 84.83 / 84.93 | 72.4 / 61.25 | 4.93 / 8.0 | 44.7 / 43.93 |
| **Node Relabeling** | | | | |
| CoreNumber | **89.1** / **86.06** | 75.87 / **70.99** | 8.47 / 11.3 | 54.97 / 53.4 |
| Degree | 84.83 / 82.53 | 76.0 / 69.39 | 7.27 / 11.8 | 58.23 / 56.57 |
| PageRank | 84.77 / 81.13 | 73.93 / 67.59 | 8.63 / 12.84 | 58.1 / 47.4 |
| LG{CoreNumber} | 76.27 / 76.93 | 75.47 / 65.32 | 8.77 / 12.84 | 50.9 / 52.37 |
| LG{Degree} | 87.23 / 84.46 | **76.63** / 68.66 | 8.47 / **14.67** | 48.83 / 46.3 |
| LG{PageRank} | 86.23 / 84.86 | 76.43 / 63.32 | **11.07** / 13.7 | 47.2 / 41.1 |
| **Baseline** | 83.24 / 83.04 | 69.69 / 59.81 | 4.52 / 7.72 | 41.57 / 42.5 |

Although scaling from LLaMA 8B to 70B yields substantial gains in tasks such as node counting and motif shape classification, Table 6 demonstrates that Qwen 2.5 14B—despite having fewer parameters than LLaMA 70B—achieves competitive, and in some cases superior, performance across several tasks. This is particularly evident in motif shape classification and diameter estimation, where Qwen's results rival or exceed those of the larger model. Nonetheless, in line with trends observed across LLaMA variants, diameter estimation remains a consistently challenging task, with overall accuracy remaining low regardless of model architecture or scale.

# B    PERFORMANCE AND COMPLEXITY

All considered graph measures, such as core number, degree, and PageRank, are computationally efficient. For graphs where the number of edges exceeds the number of nodes, the computational complexity scales linearly with the number of edges, that is, $\mathcal{O}(m)$, where $m$ denotes the number of edges. Given that the graphs in the evaluated datasets are relatively small, the computation time required for these measures is negligible compared to the response generation time of the LLMs.

Table 6: Accuracy scores for a subset of tasks on the **GraphWave** dataset using **Qwen 2.5 14B - 1M** as an ablation study. Notations remain the same as in Table 1.

|  | **Node Counting** | **Node Degree** | **Diameter** | **Motifs' Shape** |
|---|---|---|---|---|
| **Random Labels** | | | | |
| CoreNumber | 70.67 / 72.12 | 76.5 / 68.66 | 12.1 / 12.5 | 58.5 / 56.3 |
| Degree | 71.23 / 70.96 | 76.97 / 66.66 | 12.07 / 13.3 | 62.27 / 59.43 |
| PageRank | 69.77 / 66.99 | 79.53 / 72.09 | 12.77 / 12.04 | 56.9 / 58.5 |
| LG{CoreNumber} | 71.13 / 68.92 | 72.17 / 54.08 | 12.63 / 14.17 | 59.17 / **61.97** |
| LG{Degree} | 71.77 / 72.86 | 74.8 / 62.49 | 13.97 / **15.37** | 60.5 / 60.6 |
| LG{PageRank} | 76.53 / 76.59 | 72.07 / 65.19 | 13.07 / 12.44 | 51.9 / 52.77 |
| **Node Relabeling** | | | | |
| CoreNumber | **86.73** / **87.66** | **80.23** / **75.49** | 12.93 / 14.5 | 58.03 / 53.17 |
| Degree | 74.2 / 77.63 | 78.03 / 72.59 | 12.8 / 11.3 | 61.3 / 49.83 |
| PageRank | 73.4 / 75.23 | 79.3 / 73.66 | 14.33 / 13.14 | 59.73 / 49.83 |
| LG{CoreNumber} | 76.03 / 71.52 | 78.77 / 69.19 | 12.63 / 12.64 | **64.03** / 57.53 |
| LG{Degree} | 80.03 / 81.39 | 76.6 / 69.22 | **15.4** / 12.77 | 60.57 / 46.77 |
| LG{PageRank} | 84.0 / 84.29 | 74.8 / 67.96 | 14.47 / 12.34 | 52.53 / 44.2 |
| **Baseline** | 78.84 / 77.43 | 70.09 / 69.19 | 9.95 / 10.74 | 54.53 / 57.3 |

To illustrate this, we report on Table 7 the number of tokens generated per task along with the average time taken to produce a single response using LLama 3 8b. Numerical tasks, LLM responses are concise and rapidly converge to a final answer. In contrast, more complex tasks elicit longer responses that often involve intermediate reasoning steps.

Table 7: Tokens generated and response time for a single graph using LLaMA 3 8B across tasks.

| **Task** | **Number of Tokens** | **Inference Time (sec)** |
|---|---|---|
| Node Counting | 16 | 0.6 |
| Max Degree | 16 | 0.6 |
| Node Degree | 16 | 0.6 |
| Edge Existence | 128 | 4.8 |
| Diameter | 128 | 4.8 |
| Shortest Path | 128 | 4.8 |
| Path Existence | 128 | 4.8 |
| Motif's Shape | 16 | 0.6 |

## C  GRAPH SIZE LIMITATIONS

In our approach, the entire graph is linearized into a token sequence and embedded directly into the model's input prompt. As a result, the maximum size of the graph that can be processed in a single prompt is constrained by the model's context window. Since the sequence length is primarily determined by the number of edges, we estimate the maximum number of edges that can be encoded per prompt for each model considered in this study.

To compute these estimates, we assume that each edge requires approximately 5 tokens to represent, and that the accompanying task description consumes an average of 100 tokens. Under these assumptions, Table 8 reports the estimated edge capacity corresponding to the context window of each model.

In Table 9, we present statistics for several widely used graph datasets. Many of these graphs are sufficiently small to fit within the context window of contemporary LLMs, with notable exceptions such as large-scale social and e-commerce networks (e.g., Reddit, Amazon). This indicates that a substantial portion of benchmark graph datasets can be fully serialized and input to an LLM in a single prompt. Nevertheless, in practical applications, even graph neural networks often rely on

Table 8: Estimated Maximum Number of Edges considering Context Window Size.

| LLM | Context Length (tokens) | Max Number of Edges |
|---|---|---|
| *LLama 3 8b* | 8,192 | 1,618 |
| *LLama 3 70b* | 8,192 | 1,618 |
| *Qwen 2.5 14B 1M* | 1,010,000 | 199,980 |

sampling strategies rather than processing entire graphs at once. A similar strategy may be necessary when using LLMs for real-world graph tasks, depending on the application and scale.

Although our study primarily investigates the ability of LLMs to understand and reason over complete graph structures, we recognize that some of the tasks examined—such as node counting—are primarily diagnostic and may have limited practical relevance. These tasks are intended to serve as controlled benchmarks to assess the reasoning capabilities of LLMs, rather than to reflect typical graph processing workloads.

Table 9: Summary of graph datasets. For graph collections, average number of nodes and edges per graph are shown.

| Dataset (Source) | #Graphs | Avg. #Nodes | Avg. #Edges |
|---|---|---|---|
| AIDS (Morris et al., 2020) | 2,000 | 16 | 32 |
| MUTAG (Morris et al., 2020) | 188 | 18 | 40 |
| OGBN-Proteins (Hu et al., 2020) | 132,534 | 39 | 299 |
| Cora (Kipf & Welling, 2016) | 1 | 2,708 | 1,433 |
| Citeseer (Kipf & Welling, 2016) | 1 | 3,327 | 9,104 |
| PPI (Zitnik & Leskovec, 2017) | 24 | 2,269 | 61,318 |
| PubMed (Kipf & Welling, 2016) | 1 | 19,717 | 88,648 |
| Amazon Computers (Yang et al., 2018) | 1 | 12,752 | 491,722 |
| Reddit (Hamilton et al., 2017) | 1 | 232,965 | 114,615,892 |
| Amazon Products (Chiang et al., 2019) | 1 | 1,569,960 | 264,339,468 |

## D    PROMPT TEMPLATES

**Node Counting**
In an undirected graph $G$, $(i, j)$ means that node $i$ and node $j$ are connected with an undirected edge.
**Q:** How many nodes are in $G$?
**G:** {linearized graph}

**Max Degree**
In an undirected graph, $(i, j)$ means that node $i$ and node $j$ are connected with an undirected edge. The degree of a node is the number of edges connected to the node. Given a graph $G$ and its list of edges, respond to the following question:
**Q:** Without any justification, what is the maximum node degree in the following graph $G$?
**G:** {linearized graph}

**Node Degree**
In an undirected graph, $(i, j)$ means that node $i$ and node $j$ are connected with an undirected edge. The degree of a node is the number of edges connected to the node. Given a graph $G$ and its list of edges, respond to the following question:
**Q:** Without any justification, what is the degree of node {node} in the following graph $G$?
**G:** {linearized graph}

**Edge Existence**
In an undirected graph, $(i, j)$ means that node $i$ and node $j$ are connected with an undirected edge. Given a graph $G$ and its list of edges, respond to the following question:

**Q:** Does an undirected edge ($\{node1\}, \{node2\}$) exist in the following graph $G$?.
**G:** {linearized graph}

**Diameter**
In an undirected graph, $(i, j)$ means that node $i$ and node $j$ are connected with an undirected edge. The diameter of a graph is the length of the shortest path between the most distanced nodes. Given a graph $G$ and its list of edges, respond to the following question:
**Q:** Without any justification, what is the diameter of the following graph $G$?
**G:** {linearized graph}

**Shortest Path**
In an undirected graph, $(i, j)$ means that node $i$ and node $j$ are connected with an undirected edge. Given a graph $G$ and its list of edges, respond to the following question:
**Q:** Without any justification, what is the length of the shortest path from node $\{node1\}$ to node $\{node2\}$? If no path exists, the response is '0'.
**G:** {linearized graph}

**Path Existence**
In an undirected graph, $(i, j)$ means that node $i$ and node $j$ are connected with an undirected edge. Given a graph $G$ and its list of edges, respond to the following question:
**Q:** Does a path that connects node $\{node1\}$ and $\{node2\}$ exist in the following graph $G$?
**G:** {linearized graph}

**Motifs' Shape Classification:**
In an undirected graph, $(i, j)$ means that node $i$ and node $j$ are connected with an undirected edge. The graph contains a motif graph with strictly one of the following structures. {structure}: {definition}
**Q:** Which of the defined structures is included in the following graph?
**graph:** {linearized graph}

**MUTAG Classification:**
In an undirected graph, $(i, j)$ means that node i and node j are connected with an undirected edge.
The graph represents a chemical compound, where nodes are atoms and edges are bonds.

Given a graph G as a list of edges, respond with 0 if the compound does not have a mutagenic effect on *Salmonella typhimurium*, or 1 if the compound has a mutagenic effect on *Salmonella typhimurium*. Do not provide any explanation or justification, just output the predicted class.

Q: What is the mutagenicity class of the compound represented by the following graph? **graph:** {linearized graph}

**AIDS Classification:**
In an undirected graph, $(i, j)$ means that node i and node j are connected with an undirected edge.
The graph represents a chemical compound, where nodes are atoms and edges are bonds.

Given a graph G as a list of edges, respond with 0 if the compound does not show evidence of anti-HIV activity, or 1 if the compound shows evidence of anti-HIV activity. Do not provide any explanation or justification, just output the predicted class.

Q: What is the anti-HIV class of the compound represented by the

```
following graph?
graph: {linearized graph}
```

