# OpenReview forum: "Graph Linearization Methods for Reasoning on Graphs with Large Language Models"
_ICLR.cc/2026/Conference — Submitted to ICLR 2026_

### Official Review · Reviewer_hELp · 2025-10-23

**Soundness:** 2
**Presentation:** 1
**Contribution:** 1
**Rating:** 2
**Confidence:** 5

**Summary:**

The paper explores how to make large language models (LLMs) reason over graph-structured data by linearizing graphs into token sequences.

**Strengths:**

1. This work uses standard graph measures (degree, PageRank, core number) and open LLMs, making replication easy.

2. The paper is generally well-organized, with results tables that clearly compare methods and baselines.

**Weaknesses:**

1. The method amounts to a prompt-engineering heuristic, without theoretical or architectural innovation.

2. The methodology section (Section 3) is verbose and unclear, lacking pseudocode, diagrams, or step-by-step illustration of the algorithm.

3. I am quite concerned about this line of research, namely the use of LLMs for graph-related tasks. It is evident that the scalability of such methods is very limited. Real-world networks typically contain thousands of nodes, which are nearly impossible to represent effectively in natural language.

4. The evaluated networks are extremely small. Frankly, I fail to see the significance of improving LLMs on these basic graph tasks—it seems to be a direction lacking real substance.
(1) LLMs are not inherently designed for graph reasoning, and encoding graphs into natural language appears to use LLMs merely for the sake of doing so.
(2) The scalability is poor.
(3) The problems addressed are trivial, and there is no clear pathway to applying these methods to real-world, complex scenarios.

**Questions:**

1. How well would the proposed method scale to larger or real-world graphs (e.g., thousands of nodes or heterogeneous edge types)?

2. Can this proposed method address traditional graph-related problems like community detection or influence maximization?

---

### Official Review · Reviewer_U9S3 · 2025-10-28

**Soundness:** 1
**Presentation:** 1
**Contribution:** 2
**Rating:** 2
**Confidence:** 5

**Summary:**

The work focuses on the graph linearization problem and explores how to properly assign node IDs in graphs. It considers that the linearization should preserve meaningful properties such as local dependency and global alignment, and develops several methods based on graph centrality and degeneracy. In the experiments, the proposed methods are tested on several synthetic graphs and two datasets from TUDataset: AIDS and MUTAG. Results show the effectiveness of the method.

**Strengths:**

The paper focuses on an interesting direction—encoding graphs for LLMs. While many studies have explored this in specific domains such as molecules, general methods that work across different types of graphs remain relatively underexplored and deserve further investigation.

The idea of considering both local dependency and global alignment is important.

**Weaknesses:**

## The evaluation should be extended to real-world data

Testing on synthetic data is useful but not sufficient. The current evaluation is quite limited and provides little insight into practical applications. More real-world datasets should be considered, including but not limited to:

1. Social networks at different scales. Social networks can range from hundreds to millions of nodes. It is important to consider how the linearized graph size affects the context window of LLMs, while still preserving key global properties.

2. Heterogeneous networks with diverse node and edge features. These networks may include numerical and textual attributes, and their relationships can be more complex. Evaluating on such data would test the method’s robustness and generality.

3. Molecular graphs with domain constraints. Molecular structures follow strict chemical rules, and there already exist domain-specific linearization methods such as SMILES and SELFIES. The paper should compare against these approaches to demonstrate the advantages of the proposed method.

## Lack of strong baselines

The current experiments mainly compare against random ordering, which is a weak baseline and insufficient to support strong conclusions. Including more competitive and meaningful baselines would make the results more convincing.

## Methodology

The discussion on how the proposed method preserves local and global properties is important.

**Questions:**

1. Are there any suggestions on what strategies to use for node labeling in different scenarios?

2. How does the current graph linearization approach compare to image-based graph representations for LLMs?

3. How does the proposed method compare to other linearization techniques mentioned in the related work section (e.g., lines 165–179)?

---

### Official Review · Reviewer_E6a1 · 2025-10-30

**Soundness:** 2
**Presentation:** 2
**Contribution:** 2
**Rating:** 4
**Confidence:** 3

**Summary:**

1. This paper addresses the challenge of enabling Large Language Models (LLMs) to effectively reason about graph-structured data by investigating how to best convert graphs into linear sequences of tokens—a process termed "graph linearization." The core problem is that graphs are inherently non-sequential, while LLMs are trained on vast amounts of linear text. The authors argue that for LLMs to understand graphs, the linearized sequences should mimic properties of natural language, specifically ​local dependency​ (where structurally related parts of the graph appear close together in the sequence) and ​global alignment​ (where important nodes are consistently assigned the same identifiers across different graphs).

2. The main contribution is the development and evaluation of several structured graph linearization methods. These methods order the edges in a graph's list based on node importance measures—such as ​degree, ​PageRank, and ​core number—to instill local dependency. This process is further enhanced by ​node relabeling, which reassigns node identifiers based on their structural importance (e.g., the highest-degree node is always labeled "0") to achieve global alignment. The authors also explore an ​edge-centric​ variant by applying these measures to the graph's line graph. These approaches are designed as input-level prompt engineering techniques, requiring no fine-tuning or modification of the underlying LLM.

3. The proposed methods are evaluated on synthetic graph reasoning tasks (GraphWave, GraphQA) and real-world molecular graph classification datasets (AIDS, MUTAG). Experiments using models like LLaMA 3 and Qwen 2.5 show that these structured linearizations consistently and significantly outperform a baseline of random edge ordering. Key findings confirm that ordering matters, node relabeling provides an additional boost, and the best strategy can be task-specific. The work demonstrates that embedding graph structural biases directly into the prompt is a simple yet powerful way to unlock LLMs' latent capabilities for graph reasoning, offering a path toward integrating graph learning into the multimodal LLM ecosystem without architectural changes or training.

**Strengths:**

1. This paper's principal strength lies in its clear, focused, and well-motivated conceptual contribution. It successfully identifies and articulates a fundamental, under-explored problem at the intersection of LLMs and graph learning: the critical importance of the sequence orderwhen linearizing a graph for LLM consumption. The proposal of the two guiding principles—local dependency​ and ​global alignment—is both intuitive and grounded in the distributional properties of natural language, providing a strong theoretical rationale for the methods. The approach is elegantly simple and practical, as it operates purely at the input level through prompt engineering, requiring no computationally expensive fine-tuning or modifications to the underlying LLM architecture. This makes the techniques immediately applicable to a wide range of existing models and tasks, lowering the barrier to entry for leveraging LLMs on graph problems.

2. A further significant strength is the paper's rigorous and comprehensive experimental evaluation. The authors move beyond a single benchmark, validating their methods on both synthetic datasets (GraphWave, GraphQA) designed to test specific reasoning skills and real-world molecular classification tasks (AIDS, MUTAG). This multi-faceted approach convincingly demonstrates the generality and practical utility of the proposed linearizations. The systematic ablation—comparing different centrality measures, node relabeling, and edge-centric approaches against a strong random baseline—provides clear evidence for the core claims, showing that structured ordering consistently improves performance. The inclusion of results from different model families (LLaMA and Qwen) and scales also strengthens the findings, suggesting the benefits are not limited to a specific architecture. Finally, the discussion of limitations, such as context window constraints, and the commitment to reproducibility add to the work's credibility and value to the research community.

**Weaknesses:**

1. A primary weakness of this work is its insufficient engagement with the most relevant and powerful baselines, critically undermining its claim of contribution. The paper convincingly demonstrates that its structured linearizations outperform a randomordering of edges. However, in the realm of graph reasoning, the more meaningful comparison is against specialized Graph Neural Networks (GNNs) and graph transformers (e.g., Graphormer), which are architecturally designed to capture graph structure. The authors briefly mention GNNs in the real-world dataset experiments but relegate them to a simple, non-optimized baseline ("a basic GCN" with potentially omitted node features). The central claim—that input-level linearization is a viable path for graph reasoning—remains unproven without a rigorous comparison against state-of-the-art graph models that would establish a true performance ceiling. The paper does not show that its method is competitive with, let alone superior to, dedicated graph architectures, which makes its practical significance questionable.

2. Furthermore, the experimental design has significant limitations that cast doubt on the robustness of the findings. The synthetic tasks, while useful for diagnostics, are often trivial (e.g., node counting, edge existence) and may not reflect the complexities of real-world graph understanding. More critically, the paper fails to directly and convincingly test its own core hypotheses. The principle of "local dependency" is not quantitatively verified; for instance, there is no analysis measuring whether related edges are actually closer in the token sequence or if this proximity correlates with performance gains. The evaluation is solely based on end-task accuracy, leaving the proposed mechanisms as unvalidated post-hoc explanations. Similarly, the benefits of "global alignment" through node relabeling are presented anecdotally, without ablation studies that isolate its effect from the ordering strategy itself.

3. The paper's scope is also narrowly focused on a specific, and arguably less impactful, subset of graph tasks. The chosen tasks are predominantly transductive(reasoning about a given graph's explicit structure) rather than inductive(learning generalizable patterns from multiple graphs). The most compelling applications of graphs, however, often involve complex node/edge features and require generalization to unseen structures. The proposed method, which serializes the entire graph structure into the prompt, is fundamentally constrained by the LLM's context window and is impractical for anything but small graphs. The work acknowledges this limitation but does not explore how its principles could scale, for example, through sampling or hierarchical linearization. Consequently, the approach feels more like a synthetic benchmark curiosity than a scalable solution for real-world graph machine learning.

4. Finally, the conceptual framing, while intuitive, lacks depth and novelty. The ideas of using centrality measures for ordering and relabeling nodes are well-established concepts in graph algorithmics and have been used in other contexts for graph representation. The presentation of these ideas as the novel principles of "local dependency" and "global alignment" feels like a re-branding of existing concepts rather than a foundational theoretical contribution. The connection to natural language properties, though appealing, remains superficial and is not backed by a rigorous analysis of token distributions or attention patterns in the LLM. Given the lack of comparison to strong baselines, limited task scope, and the failure to empirically validate its core hypotheses, the paper's contributions appear incremental and insufficient for a top-tier conference publication.

**Questions:**

See above.

---

### Official Review · Reviewer_5sN4 · 2025-11-01

**Soundness:** 2
**Presentation:** 3
**Contribution:** 2
**Rating:** 2
**Confidence:** 4

**Summary:**

This paper examines how the ordering of graph representations affects LLM-based graph reasoning. The authors highlight two key properties that emerge when prompting pretrained LLMs with graph information serialized as sequences of tokens: local dependency (capturing nearby structural relations) and global alignment (preserving the overall graph topology). They further provide a systematic empirical study evaluating how different ordering strategies influence performance across graph reasoning tasks.

**Strengths:**

This paper introduces the concept of **graph linearization**, which aims to transform graph structures into linear sequences of tokens. It further proposes two guiding principles for effective linearization: local dependency and global alignment. Local dependency refers to the ability of large language models (LLMs) to predict subsequent or missing tokens within a sequence derived from a single linearized graph. The experimental results demonstrate that the proposed core/degree-based graph linearization method outperforms random baselines on average, highlighting the critical role of node ordering in effectively prompting pretrained LLMs.

**Weaknesses:**

1. **Limited Contribution and Novelty**. The contribution of this paper is limited, particularly in terms of novelty. Prior work has already demonstrated the sensitivity of token ordering in LLM-based graph reasoning (e.g., GraphQA, GraphText). Thus, the concept of graph linearization is not new. Moreover, the paper adopts classical heuristics such as Degree/Core/PageRank for node ordering, methods that have been widely used for decades, without proposing any new algorithmic contribution to improve ordering strategies.

2. **Theory–Experiment Gap**. The paper introduces two principles, local dependency and global alignment, as the theoretical foundations of its method. However, the experiments do not directly validate either principle. The evaluation merely reports accuracy improvements across ordering heuristics, which does not demonstrate that the proposed principles are the reason behind these gains. For instance, validating local dependency would require analyzing whether closer graph distances translate into closer sequence proximity during inference, yet no such metric or analysis is provided. As a result, the central claims remain unsubstantiated.

3. **Missing Key Baselines**. The experimental comparison omits critical baselines. For example, GraphText explicitly argues that naive sequence representations can cause significant performance drops, while this work claims that careful ordering improves LLM reasoning by exploiting natural-language-like structures. Including such baselines is necessary to demonstrate whether graph linearization genuinely offers advantages over alternative representations. Without this comparison, the contribution is less convincing.

4. **Figure 1 Requires Improvement**. Figure 1 does not effectively convey the motivation for graph linearization. Minor issues such as the notation “Graph G” instead of “Graph (G)” should also be corrected. More fundamentally, the figure should visually articulate why and how linearization helps LLM-based graph reasoning.

5. **Questionable Effectiveness in Node Counting Task**. Table 2 raises concerns about the robustness and applicability of the proposed methods. In particular, the random ordering baseline performs comparably or even better than the linearization-based heuristics in the zero-shot node counting task. The paper should analyze and explain the cause of this behavior; otherwise, it undermines the claimed effectiveness of graph linearization.

**Questions:**

Please see the weakness part.

---

### Meta-Review · Area_Chair_xcR7 · 2025-12-28

**Summary:**

In this paper, the authors study different graph linearization methods based on graph centrality and degeneracy for large language models (LLMs), which are further enhanced using node relabeling. Experimental results show that the proposed method outperforms random baselines.

All reviewers raise concerns regarding the manuscript, including limited technical novelty and contribution, insufficient experiments, particularly missing baselines, and limited tasks and datasets, and presentation issues. The authors did not provide any rebuttal. It is clear and unanimous that the paper needs more effort and is currently below the acceptance bar.

**Reviewer Concerns:**

There is no rebuttal, and therefore all concerns remain valid.

**Reviewer Scores:**

All scores are likely to remain.

---

### Decision · Program_Chairs · 2026-01-26

Reject